statistics/biometrics

exponential hazard, extreme value theory, semi-supercentenarian

**Author for correspondence:**
Léo R. Belzile
e-mail: leo.belzile@hec.ca

# Human mortality at extreme age

Léo R. Belzile[1], Anthony C. Davison[2], Holger Rootzén[3] and Dmitrii Zholud[3]

[1]Department of Decision Sciences, HEC Montréal, 3000, chemin de la Côte-Sainte-Catherine, Montréal, Quebec, Canada H3T 2A7
[2]Institute of Mathematics, École polytechnique fédérale de Lausanne, Station 8, Lausanne 1015, Switzerland
[3]Department of Mathematical Sciences, Chalmers and Gothenburg University, Chalmers Tvärgata 3, Göteborg 41296, Sweden

LRB, 0000-0002-9135-014X; ACD, 0000-0002-8537-6191; HR, 0000-0001-8869-7989; DZ, 0000-0003-0146-6999

We use a combination of extreme value statistics, survival analysis and computer-intensive methods to analyse the mortality of Italian and French semi-supercentenarians. After accounting for the effects of the sampling frame, extreme-value modelling leads to the conclusion that constant force of mortality beyond 108 years describes the data well and there is no evidence of differences between countries and cohorts. These findings are consistent with use of a Gompertz model and with previous analysis of the International Database on Longevity and suggest that any physical upper bound for the human lifespan is so large that it is unlikely to be approached. Power calculations make it implausible that there is an upper bound below 130 years. There is no evidence of differences in survival between women and men after age 108 in the Italian data and the International Database on Longevity, but survival is lower for men in the French data.

## 1. Introduction

Solid empirical understanding of human mortality at extreme age is important as one basis for research aimed at finding a cure for ageing (described, e.g. in [1]), and is also an element in the hotly debated and societally important question whether the current increase in expected lifespan in developed countries, of about three months per year since at least 1840 [2], can continue. The limit to human lifespan, if any, also attracts considerable media attention (e.g. [3–5]).

Einmahl *et al.* [6] analyse data on mortality in The Netherlands and conclude that 'there indeed is a finite upper limit to the lifespan' for both men and women. Their dataset, provided by Statistics Netherlands and consisting of about 285 000 'Dutch residents, born in the Netherlands, who died in the years 1986–

**Figure 1.** Modified Lexis diagram of the ISTAT data, showing age and calendar time at death for men (red crosses) and women (grey dot if only one woman, black dot if several). Each individual ages along a trajectory of unit slope (not shown for clarity) that may enter the sampling frame either along its left side (if aged over 105 years on 1 January 2009, the start of the sampling frame) or its lower edge (if they reach age 105 from 1 January 2009 to 31 December 2015) and then terminates either in their death or, if they die after 31 December 2015, by censoring on that date. Thus the sample includes only persons aged at least 105 years and alive on 1 January 2009, or whose 105th birthday occurred from 1 January 2009 to 31 December 2015. Censored observations are displayed in the right margin; the ticks indicate the ages of women (black) and men (red) alive on 31 December 2015. The death counts per sex for each calendar year are given at the top of the graph.

2015 at a minimum age of 92 years', had not undergone any validation procedure. As might be expected, the vast majority died before their 100th birthdays: 99.5% lived 107 or fewer years, and 97% died at age 101 or younger. The cohorts for analysis were taken to be the calendar years of death, and truncation of lifetimes was not taken into account. Hanayama & Sibuya [7] also find an upper lifespan limit of 123 years for Japanese persons aged 100 or more, by fitting a generalized Pareto distribution to 1-year and 4-year birth cohorts, taking into account the sampling scheme. In both cases, any plateauing of mortality may be confounded with the increase in hazard between ages 100 and 105, and this would invalidate the extrapolation to yet higher ages.

The validity of conclusions about mortality at extreme age depends crucially on the quality of the data on which they are based [8], as age misrepresentation for the very old is common even in countries with otherwise reliable statistical data [9]. Motivated by this, demographic researchers from 13 countries contribute to the International Database on Longevity (IDL), the third (August 2021) release of which contained 1119 individually validated life lengths of supercentenarians, i.e. those reaching age 110 or more; the data, which cover different time periods for different countries, can be obtained from www.supercentenarians.org. For some countries, the IDL now also includes data on semi-supercentenarians, i.e. people living to an age of at least 105. Since October 2019, IDL has contained French data on 9571 semi-supercentenarians and 241 supercentenarians who died between 1 January 1987 and 31 December 2016. We call these the France 2019 data; all these supercentenarians but only some of the semi-supercentenarians were validated.

An earlier release of the IDL was analysed by Gampe [10] and Rootzén & Zholud [11], the latter with extensive discussion [12]. Both papers made allowance for the sampling scheme, and in particular for the truncation of lifetimes that it entails. They concluded that there is no indication of an increase in mortality for ages above 110 years, and hence no indication of a finite upper limit to the human lifespan. Rootzén & Zholud [11] also found no differences in mortality between men and women or between populations from regions and countries as varied as Japan, the USA or Europe. These conclusions are striking, but the small size of that release of the IDL and the lack of balance between the subgroups limited the statistical power available to detect such differences.

The Italian National Institute of Statistics (ISTAT) has recently produced an important new database containing individually validated birth dates and survival times in days of all persons in Italy who were at least 105 years old at some point from 1 January 2009 to 31 December 2015. Using advanced survival analysis tools, Barbi *et al.* [13] found that death rates in this dataset 'reach or closely approach a plateau after age 105' and found a small but statistically significant cohort effect.

Data analysis must take into account the sampling scheme underlying such data. The ISTAT lifetimes are left-truncated because only individuals who attain an age of at least 105 years during the sampling period are included, and they are right-censored because the date of death of persons still alive in 2016 is unknown; see figure 1. The right-censored lifetimes, shown by the tick marks at the right side

of the panel, include the oldest individual; ignoring either the truncation or the censoring could lead to incorrect conclusions. The France 2019 lifetimes are left- and right-truncated: only individuals who are observed to die during the sampling period appear in the dataset. The statistical consequences are discussed in appendix A.2.

In our analysis, we take the sampling frame into account, pinpoint the age, if any, at which mortality attains a plateau, and disentangle the effects of age and of birth cohort. We also compare mortality in the ISTAT, France 2019 and IDL data, and between men and women.

We use the generalized Pareto distribution from extreme value statistics in the main analysis, supplemented by fits of the Gompertz distribution, which is standard in demography. We first outline our main results and conclusions; the appendix gives a more detailed description of our methods.

## 2. Results for ISTAT data

Lifetimes beyond 105 years are highly unusual and the application of extreme value models [14] is warranted. We use the generalized Pareto distribution,

$$F(x) = \begin{cases} 1 - (1 + \gamma x/\sigma)_+^{-1/\gamma}, & x \geq 0, \gamma \neq 0, \\ \exp(-x/\sigma), & x \geq 0, \gamma = 0, \end{cases} \tag{2.1}$$

to model $x$, the excess lifetime above $u$ years. In equation (2.1), $a_+ = \max(a, 0)$ and $\sigma > 0$ and $\gamma \in \mathbb{R}$ are scale and shape parameters. For negative shape parameter $\gamma$, the random variable has a finite upper endpoint at $-\sigma/\gamma$, whereas $\gamma \geq 0$ yields an infinite upper endpoint.

The corresponding hazard function, often called the 'force of mortality' in demography, is the density evaluated at excess age $x$, conditional on survival to then, i.e.

$$h(x) = \frac{f(x)}{1 - F(x)} = \frac{1}{(\sigma + \gamma x)_+}, \quad x \geq 0, \tag{2.2}$$

where $f(x) = \mathrm{d}F(x)/\mathrm{d}x$ is the generalized Pareto density function. If $\gamma < 0$, the hazard function tends to infinity at the finite upper limit for exceedances. When $\gamma = 0$, $F$ is exponential and the hazard function is constant, meaning that the likelihood that a living individual dies does not depend on age beyond the threshold. If so, the force of mortality is said to have plateaued at age $u$.

The choice of a threshold $u$ above which equation (2.1) models exceedances appropriately is a basic problem in extreme value statistics and is surveyed by Scarrott & MacDonald [15]. If $u$ is high enough for equation (2.1) to provide an adequate approximation to the distribution of exceedances, then the shape parameter $\gamma$ is approximately unchanged if a higher threshold $u'$ is used, and the scale parameters for $u$ and $u'$ have a known relationship, so a simple and commonly used approach to the choice of threshold is to plot the parameters of the fitted distributions for a range of thresholds [16] and to use the lowest threshold above which parameter estimates stabilize. This choice balances the extrapolation bias arising if the threshold is too low with the increased variance incurred when taking $u$ too high to retain an adequate number of observations.

Figure 2a shows that for age thresholds close to 105 years the estimated shape parameters for excess life lengths are negative, with 95% confidence intervals barely touching zero, but there is no systematic indication of non-zero shape above 107 years. Figure 2b displays the estimated scale parameter of the exponential model fitted to life lengths exceeding the threshold. The scale parameters decrease for ages 105–107 but show no indication of change after age 107, where the scale parameter estimate is 1.45. Parameter stability plots suggest an exponential model and hence a constant hazard after age 107 or so, where a mortality plateau seems to be attained. A more formal analysis supporting such a threshold is given in appendix A.3.

The upper part of table 1 shows results from fitting equation (2.1) and the exponential distribution to the ISTAT data for a range of thresholds. The exponential model provides an adequate fit to the exceedances over a threshold at 108 years, above which the hypothesis that $\gamma = 0$, i.e. the exponential model is an adequate simplification, is not rejected. Diagnostic plots are shown in figure 6 of the appendix.

The estimated scale parameter obtained by fitting an exponential distribution to the ISTAT data for people older than 108 is 1.45 (years) with 95% confidence interval (1.29, 1.61). Hence the hazard is estimated to be 0.69 (years$^{-1}$) with 95% confidence interval (0.62, 0.77); above 108 years the estimated probability of surviving at least one more year at any given age is 0.5 with 95% confidence interval (0.46, 0.54).

We investigated effects of year of birth, but found none; see appendix A.4.

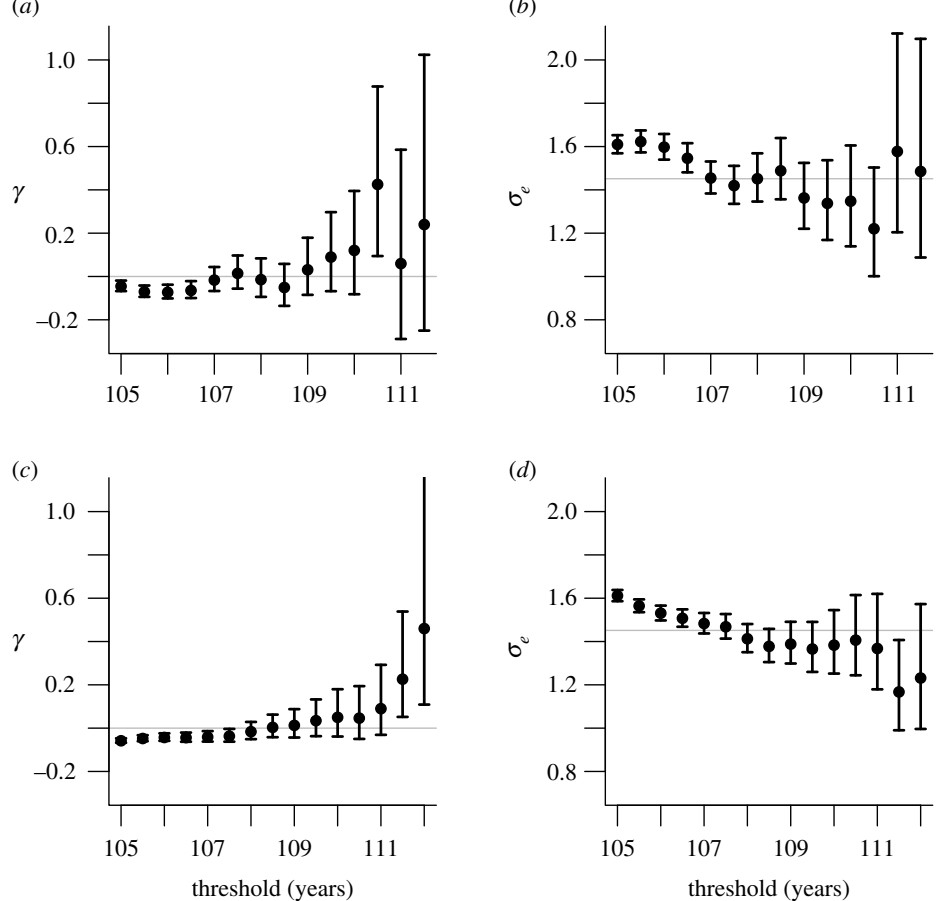

**Figure 2.** Parameter stability plots for the ISTAT data (top) and for the France 2019 data (bottom), showing the shape $\gamma$ of the generalized Pareto distribution (left) and the scale $\sigma_e$ of the exponential distribution (right) based on lifetimes that exceed the age threshold on the *x*-axis. The plots give maximum likelihood estimates with 95% confidence intervals derived using a likelihood ratio statistic. The horizontal lines in the right-hand panels correspond to the estimated scale for excess lifetimes over 108 years for the ISTAT data.

## 3. Results for France 2019 data

Estimation for the France 2019 data was performed as described by Rootzén & Zholud [11], taking into account the left- and right-truncation of the lifetimes. The parameter stability plots in the lower panels of figure 2 show a small increase in estimated shape with the threshold; table 1 shows that there is a compensating decrease in the estimated scale parameters. This is due to the presence of Jeanne Calment: her age at death, 44 724 days, exceeds the second highest French lifetime by more than 7 years, and as the threshold increases, the influence of her lifetime on the fitted model results in larger shape estimates and increased uncertainty.

The lower part of table 1 indicates that the exponential and generalized Pareto models fit equally well above 108 years, so we prefer the more parsimonious exponential fit; see appendices A.3 and A.5 and figure 7 for more detail, including a formal check on the suitability of the chosen thresholds. For French persons older than 108, the exponential scale parameter is estimated to be 1.41 (years) with 95% confidence interval (1.32, 1.51), the exponential hazard is estimated to be 0.71 (years$^{-1}$) with 95% confidence interval (0.66, 0.76) and the estimated probability of surviving at least one more year is 0.49 with 95% confidence interval (0.47, 0.52).

Table 2 shows that estimates of the scale parameter for the exponential distribution for women and men for the France 2019 data differ. If men are excluded, then the estimated scale parameter increases from 1.41 to 1.46 years, and if Jeanne Calment is also excluded, the estimate for women drops to 1.45 years. Similarly to the ISTAT data, survival for ages 105–107 was lower in earlier cohorts.

**Table 1.** Estimates (s.e.) of scale and shape parameters ($\sigma$, $\gamma$) for the generalized Pareto model and of the scale parameter ($\sigma_e$) for the exponential model for the ISTAT and France 2019 datasets as a function of threshold, with number of threshold exceedances ($n_u$), $p$-value for the likelihood ratio test of $\gamma = 0$ and for testing the null hypothesis $\gamma \geq 0$ (infinite upper endpoint) against the alternative $\gamma < 0$ (finite upper endpoint) based on the profile likelihood ratio test under the generalized Pareto model ($p_\infty$).

| threshold | 105 | 106 | 107 | 108 | 109 | 110 | 111 |
|---|---|---|---|---|---|---|---|
| ISTAT | | | | | | | |
| $n_u$ | 3836 | 1874 | 947 | 415 | 198 | 88 | 34 |
| $\sigma$ | 1.67 (0.04) | 1.70 (0.06) | 1.47 (0.08) | 1.47 (0.11) | 1.33 (0.15) | 1.22 (0.23) | 1.5 (0.47) |
| $\gamma$ | −0.05 (0.02) | −0.07 (0.03) | −0.02 (0.04) | −0.01 (0.06) | 0.03 (0.09) | 0.12 (0.17) | 0.06 (0.30) |
| $\sigma_e$ | 1.61 (0.03) | 1.60 (0.04) | 1.45 (0.05) | 1.45 (0.08) | 1.36 (0.11) | 1.35 (0.17) | 1.58 (0.32) |
| $p$-value | 0.04 | 0.01 | 0.70 | 0.82 | 0.74 | 0.44 | 0.84 |
| $p_\infty$ | 0.02 | 0.01 | 0.35 | 0.41 | 0.63 | 0.78 | 0.58 |
| France 2019 | | | | | | | |
| $n_u$ | 9808 | 5026 | 2471 | 1208 | 548 | 241 | 106 |
| $\sigma$ | 1.68 (0.02) | 1.58 (0.03) | 1.53 (0.04) | 1.43 (0.06) | 1.37 (0.08) | 1.33 (0.13) | 1.27 (0.18) |
| $\gamma$ | −0.06 (0.01) | −0.04 (0.01) | −0.04 (0.02) | −0.02 (0.03) | 0.01 (0.05) | 0.05 (0.08) | 0.09 (0.11) |
| $\sigma_e$ | 1.61 (0.02) | 1.53 (0.03) | 1.48 (0.03) | 1.41 (0.05) | 1.39 (0.07) | 1.38 (0.11) | 1.37 (0.16) |
| $p$-value | $8 \times 10^{-7}$ | 0.01 | 0.06 | 0.60 | 0.78 | 0.46 | 0.32 |
| $p_\infty$ | $4 \times 10^{-7}$ | $4 \times 10^{-3}$ | 0.03 | 0.30 | 0.61 | 0.77 | 0.84 |

## 4. Power

Our analysis above suggests that constant hazard adequately models the lifetimes over 108 years, and extrapolated indefinitely this would imply that there is no limit to the human lifespan. One might wonder whether an increasing hazard would be detectable, however, as the number of persons attaining such ages is relatively small. To assess this, we performed a simulation study described in appendix A.6, mimicking the sampling schemes of the ISTAT, France 2019 and IDL (without the French data, to eliminate overlap) datasets as closely as possible and generating samples from the generalized Pareto distribution with $-0.25 \leq \gamma \leq 0$.

Any biological limit to their lifespan should be common for all humans, whereas differences in mortality rates certainly arise due to social and medical environments and can be accommodated by letting hazards vary by factors such as country or sex. With the overlap dropped we can treat the datasets as independent and compute the power for a combined likelihood ratio test of $\gamma = 0$ (infinite lifetime) against alternatives with $\gamma < 0$ (finite lifetime). For concreteness of interpretation, we express the results in terms of the implied upper limit of lifetime, i.e. $\iota = u - \sigma/\gamma$. Figure 3a shows the power curves for the three datasets individually and pooled. The power of the likelihood ratio test for the alternatives $\iota \in \{125, 130, 135\}$ years, for example, is 0.45/0.32/0.24 for the ISTAT data above 108, 0.82/0.60/0.45 for the France 2019 data above 108, and 0.75/0.51/0.37 for the IDL data above 110. The power for $\iota = 125/130/135$ years based on all three datasets is 0.99/0.88/0.72, so it is implausible that any upper limit to the human lifespan is below 130 years or so.

Similar calculations give the power for testing the null hypothesis $\gamma = 0$ against alternatives $\gamma < 0$. Forcing all datasets to have the same shape parameter would allow them to have different endpoints so we reject the overall null hypothesis if we reject the exponential hypothesis, $\gamma = 0$, for any of the three datasets. The power of this procedure is also shown in figure 3. The resulting combined power exceeds 0.8 for $\gamma < -0.065$ and equals 0.97 for the alternative $\gamma = -0.09$, giving strong evidence against a sharp increase in the hazard function after 108 years.

## 5. Gompertz model

The hazard function of the generalized Pareto distribution cannot model situations in which the hazard increases to infinity but the upper limit to lifetimes is infinite. This possibility is encompassed by the

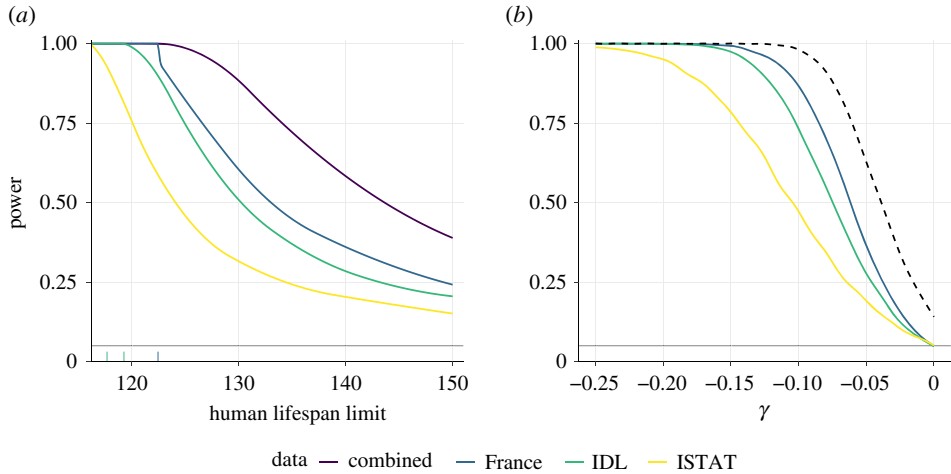

**Figure 3.** Power functions based on the IDL (excluding French records), France 2019 and ISTAT databases and combined dataset, with rugs showing the lifetimes above 115. (*a*) Power for testing the null hypothesis of infinite endpoint against the alternative of a finite endpoint *ι*, based on the likelihood ratio statistic. The endpoint cannot be below the largest lifetime in each database. (*b*) Power of the Wald statistic for testing the null hypothesis $\gamma = 0$ against the one-sided alternative $\gamma < 0$, as a function of $\gamma$; the dashed line represents the power obtained by rejecting exponentiality when any of the three one-sided tests rejects. The curves are obtained by conditioning on the birthdates and left-truncated values in the databases, then simulating generalized Pareto data whose parameters are the partial maximum likelihood estimates $(\widehat{\sigma}_\gamma, \gamma)$. The simulated records are censored if they fall outside the sampling frame for the ISTAT data and are simulated from a left- and right-truncated generalized Pareto distribution for IDL and France 2019. See appendix A.6 for more details.

**Table 2.** Estimates of the scale, $\sigma_e$, of the exponential distribution, with 95% Wald-based confidence intervals (CI). This distribution is fitted to exceedances of 108 years in the ISTAT and France 2019 data and of 110 years in the IDL data.

|  | ISTAT | | France 2019 | | IDL | |
|---|---|---|---|---|---|---|
|  | *n* | $\sigma_e$ (95% CI) | *n* | $\sigma_e$ (95% CI) | *n* | $\sigma_e$ (95% CI) |
| women | 375 | 1.45 (1.31, 1.60) | 1110 | 1.46 (1.36, 1.56) | 728 | 1.31 (1.21, 1.41) |
| men | 40 | 1.41 (0.98, 1.85) | 94 | 0.90 (0.70, 1.10) | 72 | 1.49 (1.12, 1.86) |
| all | 415 | 1.45 (1.31, 1.59) | 1204 | 1.41 (1.32, 1.51) | 800 | 1.33 (1.23, 1.42) |

Gompertz distribution [17], which has long been used for modelling lifetimes and often provides a good fit to data at lower ages (e.g. [18]). When the Gompertz model is expressed in the form

$$F(x) = 1 - \exp[-\{\exp(\beta x/\sigma) - 1\}/\beta], \quad x > 0, \ \sigma, \beta > 0,$$

$\sigma$ is a scale parameter with the dimensions of $x$, and the dimensionless parameter $\beta$ controls the shape of the distribution. Letting $\beta \to 0$ yields the exponential distribution with mean $\sigma$; small values of $\beta$ correspond to small departures from the exponential model. The fact that $\beta$ cannot be negative affects statistical comparison of the Gompertz and exponential models; see appendix A.7.

The Gompertz distribution has infinite upper limit to its support, so it cannot be used to assess whether there is a finite upper limit to the human lifespan. Its hazard function, $\sigma^{-1} \exp(\beta x/\sigma)$, is finite but increasing for all $x$ ($\beta > 0$) or constant ($\beta = 0$). The limiting distribution for threshold exceedances of Gompertz variables is exponential, and this limit is attained rather rapidly, so a good fit of the Gompertz distribution for lower $x$ would be compatible with good fit of the exponential distribution for threshold exceedances at higher values of $x$.

Computations summarized in appendix A.7 show that the exponential model, and hence also the Gompertz model with very small $\beta$, give equally good fits to the Italian and the French datasets above age 107, and that the Gompertz and generalized Pareto models fit equally well above age 105.

# 6. Conclusion

Table 2 shows no differences between survival after age 108 in the ISTAT data and survival after age 110 in the IDL data for women, for men, or for women and men combined, so we obtained combined estimates by pooling the two databases. The resulting estimates also show no significant differences in survival between men and women, and we conclude that survival times in years after age 108 in the ISTAT data and after age 110 in the IDL data are well described by an exponential distribution with scale parameter 1.37 and 95% confidence interval (1.29, 1.45). The corresponding estimated probability of surviving one more year is 0.48 with 95% confidence interval (0.46, 0.5).

There was no indication of differences in survival for women in the France 2019 data and in the combined ISTAT and IDL data, but survival for men was lower in the France 2019 data. Pooling the ISTAT data, the France 2019 data and the IDL data with France removed gives an exponential scale parameter estimate of 1.39 years with 95% confidence interval (1.33, 1.45), and estimated probability 0.49 (0.47, 0.50) of surviving one more year. Deleting the men from the France 2019 data or dropping Jeanne Calment changes estimates and confidence intervals based on these pooled exponential models by at most two units in the second decimal place. Moreover, there is no evidence that the Gompertz model, with increasing hazard, fits better than the exponential model, with constant hazard, above 108 years.

There is high power for detection of an upper limit to the human lifespan up to around 130 years, based on fits of the generalized Pareto model to the three databases. This does not mean such ages will be reached sometime soon, however, as the probability of surviving until 130 conditional on reaching 110 years approximately equals that of seeing heads on 20 consecutive tosses of a fair coin. This event has a probability of less than one in a million and is highly unlikely to occur in the near future, though the increasing number of supercentenarians makes it possible that the maximum reported age at death will rise to 130 years during the present century [19].

# 7. Discussion

The results of the analysis of the newly available ISTAT data agree strikingly well with those for the IDL supercentenarians and for the women in the France 2019 data. Once the effects of the sampling frame are taken into account by allowing for truncation and censoring of the ages at death, a model with constant hazard after age 108 fits all three datasets well; it corresponds to a constant probability of 0.49 that a living person will survive for one further year, with 95% confidence interval (0.47, 0.50). Power calculations make it implausible that there is an upper limit to the human lifespan of 130 years or below.

Although many fewer men than women reach high ages, no difference in survival between the sexes is discernible in the ISTAT and the IDL data. Survival of men after age 108 is lower in the France 2019 data, but it seems unlikely that this reflects a real difference. It seems more plausible that this is due to gender imbalance, some form of age bias or is a false positive caused by multiple testing.

If the ISTAT and France 2019 data are split by birth cohort, then we find roughly constant mortality from age 105 for those born before the end of 1905, whereas those born in 1906 and later have lower mortality for ages 105–107; this explains the cohort effects detected by [13]. Possibly the mortality plateau is reached later for later cohorts. The plausibility of this hypothesis could be weighed if further high-quality data become available.

Data accessibility. The ISTAT data can be obtained from the Italian National Institute of Statistics by registering at the Contact Center and mentioning the Semi-supercentenarian Survey and Marco Marsili (marsili@istat.it) as contact person. These data are accessible with a nominal financial payment: at the editors' request, the authors contacted ISTAT to ask for this fee to be waived, and they were reluctant to do so. Given that a named contact can be approached to access the data, the journal's editors have made a rare exception to the normal data deposition requirements. The IDL data can be obtained freely by registering at https://www.supercentenarians.org.

Authors' contributions. L.R.B., A.C.D. and H.R. wrote the paper and appendix. All authors contributed to the statistical analyses. L.R.B. wrote the R code to generate the figures and tables, and D.Z. authored the MATLAB toolbox.

Competing interests. We declare we have no competing interests.

Funding. A.C.D. is supported by the Swiss National Science Foundation.

Acknowledgements. We thank contributors to the International Database on Longevity and the Istituto Nazionale di Statistica for providing the data.

# Appendix A

Here we describe the methods used to obtain our results, produce the figures and perform our inferences, and add further information on goodness of fit and hazard estimates.

## A.1. Reproducibility

R [20] was used for all of the analyses and to generate all the tables and figures. Code to reproduce the analyses is provided in the online electronic supplementary material.

Alternative analyses may be performed using the R package longevity or the MATLAB toolbox for life-length analysis LATool. The latter consists of three files that are available on GitHub, https://github.com/OGCJN/Human-mortality-at-extreme-age.git.

The IDL data used in this article, extracted in August 2021, comprise the records of all French semi-supercentenarians and supercentenarians who died after 31 December 1986, and the records of supercentenarians from the third IDL release for Austria, Belgium, Canada (Quebec), Denmark, England and Wales, Finland, Germany, Norway, Spain, Sweden and the USA. The online electronic supplementary material also explains how to obtain these and the Italian data used in the paper, and gives code to preprocess them prior to reproducing our analyses.

## A.2. Truncation and censoring

Figure 1 shows that many persons in the ISTAT dataset were alive on 31 December 2016, when sampling finished, so their lifetimes must be treated as right-censored. Lifetimes above 105 years on 1 January 2009 are left-truncated and individuals not aged at least 105 from 1 January 2009 to 31 December 2016 are not included. Both censoring and truncation must be handled correctly to avoid biased inferences; the effect of the truncation is that inferences must be conditioned on the event that an individual appears in the database. Below we outline how this is achieved; see [11,21] for more details.

Consider a sampling interval $\mathcal{C} = (b, e)$ of calendar time during which individuals aged over a threshold of $u$ years were observed. Let $x = \text{age} - u$ denote the excess age of an individual who dies aged older than $u$, having reached age $u$ at calendar time $t$. Assume that the excess ages $x$ are independent with cumulative distribution function $F(x; \boldsymbol{\theta})$, probability density function $f(x; \boldsymbol{\theta})$, and survival function $S(x, \boldsymbol{\theta}) = 1 - F(x, \boldsymbol{\theta})$, where $\boldsymbol{\theta}$ is a vector of parameters to be estimated.

The likelihood contribution for someone who died in the interval $\mathcal{C}$ is then

$$\frac{f(x)}{S\{(b - t)_+\}}, \quad x > 0,$$

whereas that for someone who is known to be alive at the end of $\mathcal{C}$, and thus whose lifetime is censored, is

$$\frac{S(e - t)}{S\{(b - t)_+\}}, \quad t < e.$$

The likelihood function $L(\boldsymbol{\theta})$ is the product of the likelihood contributions from all individuals included in the data under study. Estimates for the generalized Pareto and Gompertz distributions were found by numerical maximization of the log-likelihood, with standard errors obtained from the inverse observed information matrix. Explicit expressions exist for the maximum-likelihood estimator of the exponential distribution scale parameter and its standard error, i.e.

$$\widehat{\sigma}_e = \frac{\sum_i \{x_i - (b - t_i)_+\}}{\#\text{deaths}}, \quad \text{se}(\widehat{\sigma}_e) = \frac{\widehat{\sigma}_e}{\sqrt{\#\text{deaths}}}. \tag{A1}$$

The IDL supercentenarian data are left- and right-truncated, and this was taken into account in our analysis; the likelihood contribution for these individuals is

$$\frac{f(x_i)}{F(e - t_i) - F\{(b - t_i)_+\}}, \quad b - t_i \leq x_i \leq e - t_i. \tag{A2}$$

Inappropriate analysis can lead to misleading results: for example, fitting an exponential distribution to the ISTAT individuals who survive beyond age 107 without accounting for truncation or censoring gives the estimate $\widehat{\sigma}_e = 1.25$ with 95% confidence interval (1.17, 1.33), to be compared with $\widehat{\sigma}_e = 1.45$

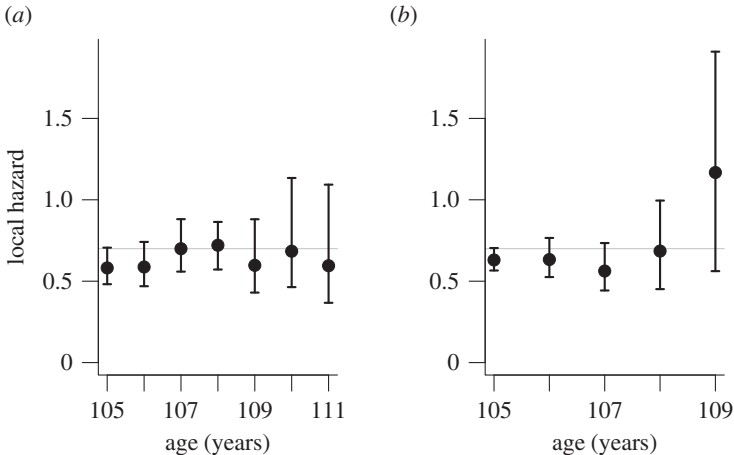

**Figure 4.** Local hazard estimates with 95% pointwise confidence intervals for ISTAT individuals born in 1886–1905 (*a*) and 1906–1910 (*b*), with horizontal lines at 0.7. The rightmost point includes all survivors beyond 111 years (respectively, 109 years).

**Table 3.** The *p*-values for the likelihood ratio test comparing the generalized Pareto and exponential models (null hypothesis) with thresholds 108 (ISTAT, France 2019) and 110 years (IDL) against the piecewise generalized Pareto model of [22] with equally spaced thresholds at the sample quintiles of the exceedances. The *p*-values are based on the asymptotic $\chi^2$ distribution.

|  | gen. Pareto | exponential |
|---|---|---|
| ISTAT | 0.47 | 0.61 |
| France 2019 | 0.44 | 0.55 |
| IDL | 0.77 | 0.77 |

with 95% confidence interval (1.35, 1.56) once the truncation and censoring are accounted for; these confidence intervals do not overlap.

## A.3. Threshold stability

For a formal assessment of threshold stability, we fit a piecewise generalized Pareto model over the $K$ regions above the thresholds $u_1 < \cdots < u_K$, each with its own shape parameter $\gamma_k$ but with scale parameters constrained to ensure that the density function is continuous at the thresholds [22]; this reduces to the usual generalized Pareto model if $\gamma_1 = \cdots = \gamma_K$. The *p*-values for testing $\gamma_1 = \cdots = \gamma_K$ against the alternative of different values of $\gamma$ with $K = 4$ thresholds at the 0, 0.2, …, 0.8 quantiles of the exceedances over 108 or 110 years are shown in table 3. They cast no doubt on the chosen thresholds for either the generalized Pareto model or with a similar test with a piecewise exponential model.

## A.4. ISTAT cohort effects

The local hazard estimates in figure 4 are obtained by splitting the likelihood contribution of individuals into yearly blocks, using disjoint intervals with $(b, e) = (a, a + 1)$ years to avoid using the same individuals several times. For the highest interval we set $e = \infty$ and include all individuals who survived into that interval.

The parameter stability plots in figure 5 and the estimated hazard plots in figure 4 show roughly constant mortality for those born in 1886–1905 for the entire age range. Mortality is lower for ages 105 and 106 for persons born in 1906–1910, but equals that for the earlier group at ages 107 and above. This reduction in mortality for the later births implies that plateauing for the entire ISTAT dataset does not start until approximately age 108 years.

## A.5. Graphical diagnostics

A quantile-quantile (or QQ-) plot is a standard diagnostic of the fit of a specified distribution to data, but it must be modified to accommodate censoring or truncation. Figure 6*a* graphs the ordered ages at death, $y_i$, of uncensored ISTAT individuals against plotting positions from $\widehat{F}^{-1}\{\widetilde{G}(y_i)\}$ [24], where $\widehat{F}^{-1}$ is the

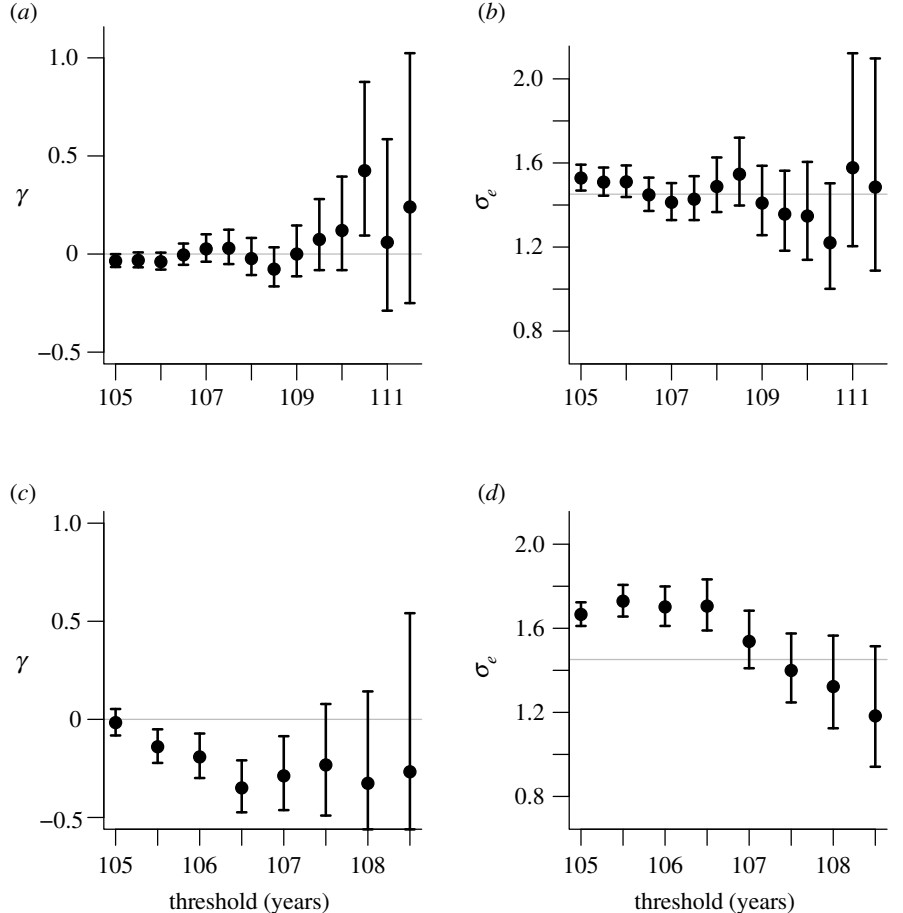

**Figure 5.** Parameter stability plots for ISTAT individuals born in 1896–1905 (*a,b*) and 1906–1910 (*c,d*), for the parameters $\gamma$ of the generalized Pareto distribution (*a,c*) and for the scale $\sigma_e$ of the exponential distribution (*c,d*) obtained from exceedances of the age threshold on the *x*-axis. The plots give maximum likelihood estimates with (profile) likelihood 95% confidence intervals. The horizontal line on the right panels corresponds to the estimated scale for exceedances above 108 for the full ISTAT dataset.

quantile function of the exponential distribution fitted to ages exceeding 108 years and $\widetilde{G}$ is the non-parametric maximum likelihood estimator of the distribution function, corrected for censoring and truncation [25,26].

To assess the variability of the plot, we simulate new ages at death from the fitted model, conditioning on the birth dates, truncation time and censoring indicator; this amounts to simulating new lifetimes from a left- and right-truncated exponential distribution, since individuals whose death is observed during the sampling frame cannot exceed the age they would reach at $c_2$, i.e. 31 December 2015 for these data. Both $\widehat{F}$ and $\widetilde{G}$ are re-estimated using the simulated samples and evaluated at a grid of fixed values $y' \in \{y_1, \ldots, y_m\}$. The approximate 95% pointwise and simultaneous simulation envelopes shown in the panel are obtained using $\widehat{F}_b^{-1}\{\widetilde{G}_b(y')\}$ ([27], §4.2.4).

For left-truncated and right-censored data, we can also compare the non-parametric, Nelson–Aalen, conditional cumulative hazard function estimate with its parametric counterparts; uncertainty assessment for the former is discussed in ([23], p. 210). QQ-plotting for left- and right-truncated observations is awkward, but the cumulative hazard can again be estimated fairly readily. Figure 6*b* and figure 7 show conditional cumulative hazard plots for the ISTAT, France 2019 and IDL data. The fits of the parametric models to the first and third of these datasets appear satisfactory, with the fit for the France 2019 data perhaps somewhat less so, though the uncertainty in the upper tail is very large.

## A.6. Power

To assess the statistical power of our procedures, we computed the maximum profile likelihood estimates $\widehat{\sigma}_\gamma$ for the original data with $\gamma$ fixed at the values $-0.25, \ldots, 0$ and simulated new excess lifetimes for each $\gamma$, conditioning on the sampling frame. In the ISTAT data, for example, we calculated the ages of all

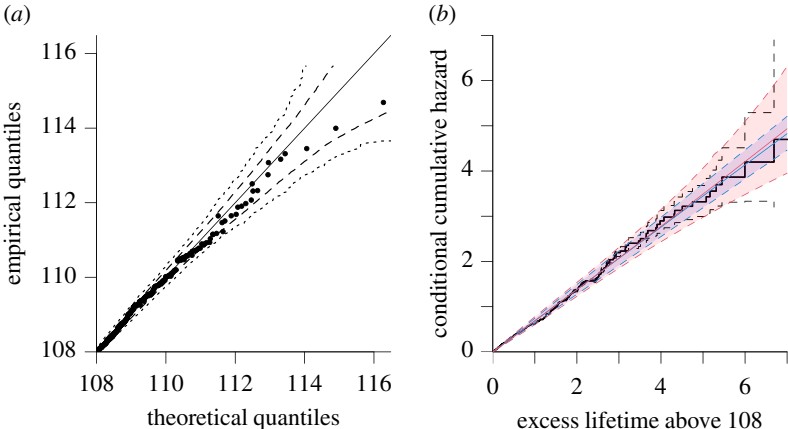

**Figure 6.** Graphical goodness-of-fit assessment for the ISTAT data. (*a*) Exponential QQ-plot of individuals who lived beyond 108 years and died before 2016, with pointwise (dashed) and simultaneous (dotted) 95% simulation envelopes. The circles contrast the ordered ages at death of non-censored ISTAT lifetimes (i.e. the observed quantiles) with the corresponding theoretical quantiles. (*b*) Non-parametric estimate of the conditional cumulative hazard function for the ISTAT data (solid black), and approximate 95% simultaneous equal precision confidence bands ([23], p. 210) (dashed black); the cumulative hazard is conditional because it applies only to individuals older than 108 years. The coloured lines and shaded bands show the maximum likelihood cumulative hazard estimates for the fitted exponential (blue) and generalized Pareto distributions (red) with pointwise 95% confidence intervals.

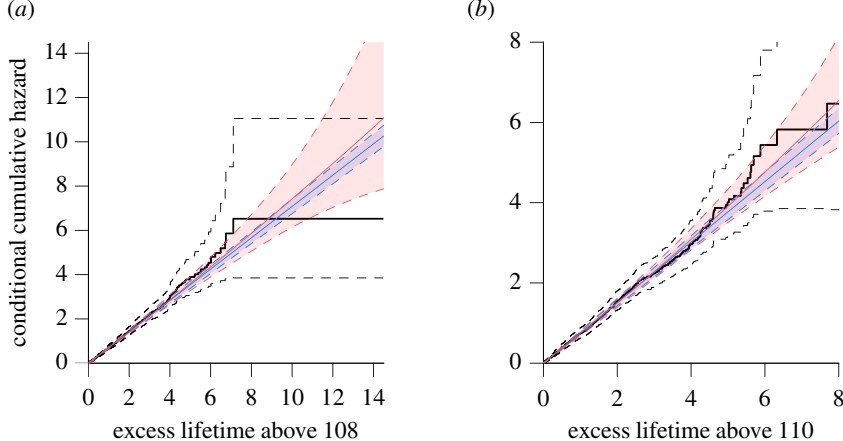

**Figure 7.** Non-parametric conditional cumulative hazard function estimates [28] for the France 2019 data above age 108 (*a*) and IDL supercentenarians (*b*) with approximate 95% equal-precision simultaneous confidence bands [29]. See caption to figure 6 for details. The non-parametric estimate for the France 2019 data is constant from 115.1 years until Jeanne Calment's death at around 122.4 years, which falls outside the pointwise envelopes for the two parametric models.

individuals on 1 January 2009, retained the dates at which they reached 108 years and sampled new excess lifetimes from the generalized Pareto distribution with parameters $(\gamma, \widehat{\sigma}_{\gamma})$, right-censoring any simulated lifetimes beyond the end of 2015. This ensures that the power calculations are as relevant as possible, not only in terms of the sampling scheme but also in terms of the underlying parameters. For each simulated dataset, we computed the directed likelihood ratio statistic $r = \mathrm{sign}(\widehat{\gamma} - \gamma)\{2\ell(\widehat{\gamma}, \widehat{\sigma}) - 2\ell(0, \widetilde{\sigma}_e)\}^{1/2}$ and the Wald statistic $w = \widehat{\gamma}/\mathrm{s.e.}(\widehat{\gamma})$ for testing the null hypothesis $\gamma = 0$ against the alternative $\gamma < 0$, which corresponds to testing for exponentiality against possible upper bounds to the human lifespan. The asymptotic null distribution of the statistics $r$ and $w$ is standard normal and we can assess the quality of this approximation by calculating the Wald statistics when $\gamma = 0$. These simulations show that the estimator of the shape parameter is unbiased and normally distributed, but the distribution of the Wald statistic is left-skewed, leading to inflated Type I error. For the power study, we thus used the simulated null distribution for comparison.

We proceeded similarly for the endpoint $\iota$. In order to simulate from generalized Pareto distributions with $\iota$ fixed at a given value, we reparametrized the log-likelihood function in terms of $\iota$ and $\sigma$ and estimated the scale parameters $\widehat{\sigma}_{\iota}$ for each of the original three datasets with $\iota$ fixed, then used the

**Table 4.** Estimates (s.e.) of Gompertz parameters ($\sigma$, $\beta$) for the ISTAT data (top) and the France 2019 data (bottom) for different thresholds, with the number of threshold exceedances ($n_u$). The bootstrap $p$-values are for the likelihood ratio test of $\beta = 0$ against $\beta > 0$. Estimates of $\beta$ reported as zero are smaller than $10^{-7}$.

| threshold | 105 | 106 | 107 | 108 | 109 | 110 |
|---|---|---|---|---|---|---|
| **ISTAT** | | | | | | |
| $n_u$ | 3836 | 1874 | 946 | 415 | 198 | 88 |
| $\sigma$ | 1.67 (0.05) | 1.71 (0.07) | 1.48 (0.08) | 1.47 (0.12) | 1.36 (0.1) | 1.35 (0.2) |
| $\beta$ | 0.05 (0.02) | 0.09 (0.04) | 0.02 (0.05) | 0.02 (0.07) | 0 | 0 |
| $p$-value | 0.02 | 0.01 | 0.37 | 0.45 | 1 | 1 |
| **France 2019** | | | | | | |
| $n_u$ | 9808 | 5026 | 2471 | 1208 | 548 | 241 |
| $\sigma$ | 1.7 (0.03) | 1.59 (0.03) | 1.54 (0.05) | 1.43 (0.06) | 1.39 (0.1) | 1.38 (0.1) |
| $\beta$ | 0.07 (0.01) | 0.05 (0.02) | 0.05 (0.03) | 0.02 (0.04) | 0 | 0 |
| $p$-value | 0 | 0 | 0.02 | 0.31 | 1 | 1 |

relation $\iota = u - \sigma/\gamma$ to obtain the three implied shape parameters $\widehat{\gamma}_\iota$. We then simulated new datasets with the sampling scheme described above, but using the three sets of parameter pairs ($\widehat{\gamma}_\iota$, $\widehat{\sigma}_\iota$). For the joint test of the null hypothesis of an exponential distribution, $\iota = \infty$, against the alternative of a common but finite $\iota$, we reparametrized the likelihood in terms of $\iota$ and allowed the three datasets to have different values of $\sigma$.

Figure 3$a$ shows how the empirical proportion of rejections for a test of nominal size 5% based on the directed likelihood root statistic $r = -\{2\ell_{\mathrm{gp}}(\widehat{\iota}, \widehat{\boldsymbol{\sigma}}) - 2\ell_{\exp}(\widetilde{\boldsymbol{\sigma}}_e)\}^{1/2}$ varies with $\iota = u - \sigma/\gamma$ for $\gamma < 0$, for the ISTAT, France 2019 and the rest of the IDL data. Here ($\widehat{\iota}$, $\widehat{\boldsymbol{\sigma}}$) are the maximum likelihood estimates for the generalized Pareto distribution parametrized in terms of a common finite $\iota$ and three scale parameters and $\widetilde{\boldsymbol{\sigma}}_e$ are the maximum likelihood estimates of the three exponential scale parameters under the null hypothesis.

## A.7. Gompertz model

The reciprocal hazard function of the Gompertz distribution $r(x) = \sigma \exp(-\beta x/\sigma)$ encodes the speed of convergence to the limiting extreme value distribution [30]; even if $\beta > 0$, $r'(x) = \beta \exp(-\beta x/\sigma) \to 0$ exponentially fast as $x \to \infty$. Any improvement in the fit of the Gompertz model for exceedances over some threshold would be shown by evidence that $\beta > 0$. This distribution places a point mass of $\exp(1/\beta)$ at $x = \infty$ when $\beta < 0$, so we allow only $\beta \geq 0$.

Table 4 summarizes the fit of this model for various thresholds and the ISTAT data, without sex or cohort effects. The hypothesis that the Gompertz distribution ($\beta > 0$) reduces to an exponential distribution ($\beta = 0$) is a boundary hypothesis, so the likelihood ratio statistic to test $\beta = 0$ does not have the usual approximate $\chi_1^2$ distribution; its large-sample distribution is a 50 : 50 mixture of a point mass at 0 and a $\chi_1^2$ distribution, sometimes written as $\frac{1}{2}\chi_0^2 + \frac{1}{2}\chi_1^2$ [31]. Barbi *et al.* [13] do not notice this, leading them to mis-state the significance of the difference of log-likelihoods in their table 2—the likelihood ratio statistic for testing $\beta = 0$ against $\beta > 0$ is $w = 2 \times 0.292 = 0.584$, and $\Pr(\chi_1^2 > 0.584) \approx 0.44$, which is the significance level quoted in [13]. In fact, the correct (asymptotic) level would be $\frac{1}{2}\Pr(\chi_1^2 > 0.584) \approx 0.22$. Here the conclusion does not change, but it might in other contexts.

In general, $p$-values obtained using a parametric bootstrap are more reliable than asymptotic approximations such as $\frac{1}{2}\chi_0^2 + \frac{1}{2}\chi_1^2$ and are preferable for such comparisons. In the present case, this entails simulating independent datasets like the original data from the boundary exponential distribution (the null hypothesis, $\beta = 0$), and estimating the $p$-value by the empirical proportion of these simulated datasets in which the likelihood ratio statistics $w^*$ are no smaller than the original value $w$, i.e. $\widehat{\Pr}_0^*(w^* \geq w)$, where the asterisk indicates a parametric bootstrap simulation and the subscript indicates that the simulation is under the null hypothesis; see ch. 4 of [27]. This approach was used to obtain the $p$-values in table 4, which show that the exponential model is statistically

**Table 5.** Deviances for comparison of extended generalized Pareto (GP) and the exponential, GP and Gompertz sub-models for different thresholds for the ISTAT data. The rows show the likelihood ratio statistic, i.e. twice the difference in log-likelihood between the specified model and an encompassing extended generalized Pareto model.

| threshold | 105 | 106 | 107 | 108 | 109 | 110 |
|---|---|---|---|---|---|---|
| Gompertz | 0.01 | 1.65 | 2.21 | 0.22 | 0.68 | 1.65 |
| GP | 0.00 | 2.17 | 2.21 | 0.23 | 0.56 | 1.05 |
| exponential | 4.16 | 8.21 | 2.36 | 0.28 | 0.68 | 1.65 |

indistinguishable from the Gompertz model from 108 years onwards, though the Gompertz model fits better at 106 years and below for the ISTAT data and at 107 years and below for the France 2019 data.

Table 5 compares the fits of the Gompertz, generalized Pareto and exponential models to the ISTAT data, with the baseline taken to be an extended generalized Pareto distribution that encompasses all three other models; the details will be reported elsewhere. The generalized Pareto and Gompertz models fit equally well for all thresholds, since the differences between their respective deviances are minimal. Both are better than the exponential model below 107 years, but not above, in agreement with table 4.

## A.8. Differences between men and women

The imbalance in the number of men and women limits our ability to detect any effects of gender on mortality at great age. To illustrate this, we conducted a simulation study based on the ISTAT data to assess the power of a likelihood ratio test for lifetime exceedances above 108 years, for which we have 375 women and 40 men; the lifetimes of 79 women and 15 men are censored. We condition on the sampling frame and the sex of individuals and simulate new life trajectories for both men and women based an exponential distribution with relative scale differing by a ratio $\lambda > 1$, corresponding to lower hazard for women. Thus individuals who were older than 108 in 2009 survive at least beyond that age, but a simulated lifetime that extends beyond 2016 is censored. For each of 10 000 simulated samples, we computed the likelihood ratio statistic for comparison of fits to men and women separately and a combined fit; the statistic has an asymptotic $\chi_1^2$ distribution. We also considered conditioning only on the birth dates and the beginning of the sampling period to assess whether the right-censoring leads to loss of power, but the difference is negligible. Power of 80% is achieved if $\lambda \approx 1.61$, corresponding to an average survival, $\sigma_e$, of 1.85 years for women and 1.14 years for men, with the corresponding differences for powers 20% and 50% being 0.28 years and 0.50 years. The power of the corresponding test for the IDL data is expected to be similar, and, if so, the combined power of the tests in the ISTAT and IDL data for detecting this ratio would be approximately 0.96.

For the France 2019 data, a likelihood ratio test strongly rejects the hypothesis of equal hazard for women and men; this is perhaps unsurprising given that the oldest Frenchman in the database died aged 111 years and 318 days, more than a decade less than Jeanne Calment. The ratio of the estimated hazards for men and women in this dataset is approximately 1.61.

## A.9. Hazard estimates

To construct a local hazard estimate based on the limiting generalized Pareto distribution, we note that this distribution has reciprocal hazard function $r(x) = (\sigma + \gamma x)_+$, where $a_+ = \max(a, 0)$ for real $a$. A more flexible functional form is $r(x) = \{\sigma + \gamma x + g(x)\}_+$, where $g(x)$ is a smooth function of $x$ that tends to zero as $x$ increases. For exploratory purposes, we take $g(x) = \sum_{k=1}^{K} \beta_k b_k(x)$, where

$$b_k(x) = (\kappa_k - x)_+^3, \quad \kappa_1 < \cdots < \kappa_K;$$

here the $\kappa_k$ are fixed knots, and $b_k(x) = 0$ for $x \geq \kappa_k$. The resulting cubic spline function $g(x)$ has two continuous derivatives. This model has $K + 2$ parameters and corresponds to the generalized Pareto distribution for $x > \kappa_K$, but allows $r(x)$ to depart from linearity for $x \leq \kappa_K$.

Lifetime data are recorded to the nearest day and often there are ties for small $x$, so we use a discrete version of the model, with $x \in \{1, \ldots, x_{\max}\}$ days, where $x_{\max}$ corresponds to 16 years after age 105. We let $h(x) = 1/r(x)$ denote the hazard function, set $H(x) = \sum_{z=1}^{x} h(z)/365$ for compatibility with the continuous

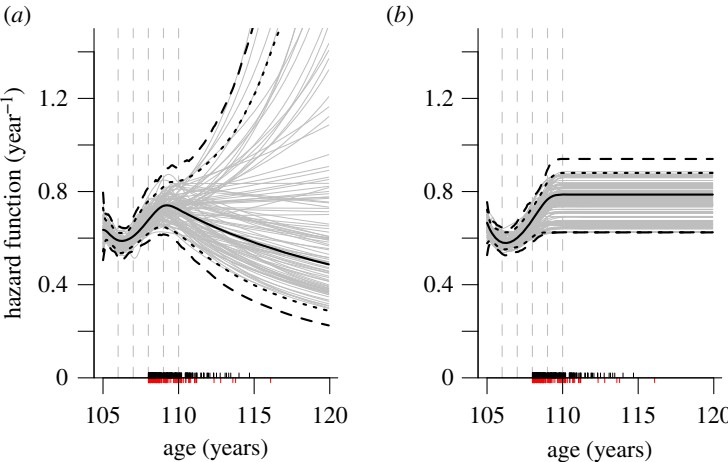

**Figure 8.** Local hazard estimates for the ISTAT data using a spline approach with five random knots centred at 106 , …, 110 years. (a) Original estimate (heavy solid line) with pointwise (dotted) and overall (dashed) 95% envelopes obtained from 5000 bootstrap replicates, of which 100 are displayed (grey solid lines). The mean positions of the knots are shown by the dashed vertical lines. The black rug shows (jittered) times of deaths, and the red rug shows (jittered) censored survival times; the rugs are suppressed for the lower ages, as there too many points to be informative. (b) The same output for the best-fitting exponential model, for which $\gamma = 0$.

case, and obtain survivor and probability mass functions $\Pr(X > x) = \exp\{-H(x)\}$ and $\Pr(X = x) = h(x)\exp\{-H(x)\}$ for $x \in \{1, …, x_{\max}\}$.

Let $X$ denote a survival time (days) beyond 105 years. For each individual, the available data are of the form $(s, d, x)$, where $s = 0$ if observation of $X$ began at 105 years and, if not, $s > 0$ is the age in days above 105 at which observation of $X$ began, $d = 1$ indicates death, $X = x$, and $d = 0$ indicates right-censoring, $X > x$. The likelihood is then a product of terms of the form

$$\frac{\Pr(X = x)^d \times \Pr(X > x)^{1-d}}{\Pr(X > s)}, \quad x > s,$$

and depends on the parameters $\sigma, \gamma, \beta_1, …, \beta_K$, which are readily estimated by numerical maximization of the log-likelihood function. The resulting fit depends on the knots $\kappa_1, …, \kappa_K$, but to reduce this dependence we generate knots at random, roughly evenly spaced in an interval $(0, x_{\max})$, where $x_{\max}$ is chosen large enough that $r(x)$ should be linear for $x > x_{\max}$, i.e. the generalized Pareto model is fitted when $x > x_{\max}$.

Figure 8a shows a local estimate of the hazard function for the ISTAT data, constrained to have the form of (2.2) above 110 years. The hazard dips up to 108 years or so, then rises and declines slowly. To assess the significance of this decline, we performed a bootstrap analysis [27,32], generating 5000 replicate datasets, the hazard function estimates for 100 of which are shown in the panel. These suggest that the slow decrease after age 110 is not significant, and this is confirmed by the pointwise and overall 95% confidence bands. The initial dip seems to be a genuine feature, but above 108 years the confidence bands cover a wide range of possible functions, including a constant hazard.

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
