## [Peer Review File · Royal Society Open Science]

Review History

RSOS-202097.R0 (Original submission)

Review form: Reviewer 1

Is the manuscript scientifically sound in its present form?

Yes

Are the interpretations and conclusions justified by the results?

Yes

Is the language acceptable?

Yes

Do you have any ethical concerns with this paper?

No

Have you any concerns about statistical analyses in this paper?

No

Recommendation?

Accept as is

Comments to the Author(s)

This manuscript is clear and convincing. It was meticulously written, with great attention to detail, organized understandably. The statistical analysis is powerful and appropriate. The manuscript is an important contribution to the literature, resolving the issue of whether human death rates level off at advanced ages or continue to increase: the rates level off.

Review form: Reviewer 2

Is the manuscript scientifically sound in its present form?

No

Are the interpretations and conclusions justified by the results?

No

Is the language acceptable?

Yes

Do you have any ethical concerns with this paper?

No

Have you any concerns about statistical analyses in this paper?

Yes

Recommendation?

Major revision is needed (please make suggestions in comments)

Comments to the Author(s)

The authors focus on lifespans of Italian and French semi-supercentenarians by taking a fully parametric approach based on the Pareto distribution. The main conclusion is that a constant force of mortality describes the data well. While this result is already known for the Italian population, it is here - and for the first time - extended to the French population. The statistical methods are generally correct and the material is nicely described. However, I have two major concerns, listed below.

MAJOR ISSUES

1. The goodness of fit of the model is overlooked. I would welcome a comparison between the cumulative hazard function predicted by the model and a Nelson-Aalen nonparametric estimate of the cumulative hazard function.
2. Lifetimes are treated as exceedances above a threshold u , which is chosen by trying several thresholds and then choosing the value that stabilize the parameter estimates. This part of the analysis looks rather empirical and it is not very convincing. The sample size decreases (and the variability of the estimates increases) as u increases and it is not clear what the authors mean by "stable estimates". I'd welcome a more rigorous approach in the selection of u .

Decision letter (RSOS-202097.R0)

Dear Professor Davison

The Editors assigned to your paper RSOS-202097 "Human mortality at extreme age" have now received comments from reviewers and would like you to revise the paper in accordance with the reviewer comments and any comments from the Editors. Please note this decision does not guarantee eventual acceptance.

Firstly, please accept our sincere apologies for the unusual delays incurred during the review process. We regret that it proved more difficult than usual to acquire referees for your paper, and Editor and staff absences related to the pandemic also caused some delays earlier in the year. We will endeavour to do all that we can to expedite your paper once you submit your revision. We invite you to respond to the comments supplied below and revise your manuscript. Below the referees' and Editors' comments (where applicable) we provide additional requirements. Final acceptance of your manuscript is dependent on these requirements being met. We provide guidance below to help you prepare your revision.

Please submit your revised manuscript and required files (see below) no later than 21 days from today's (ie 08-Jul-2021) date. Note: the ScholarOne system will 'lock' if submission of the revision is attempted 21 or more days after the deadline. If you do not think you will be able to meet this deadline please contact the editorial office immediately.

on behalf of Professor Andreas Kyprianou (Associate Editor) and Mark Chaplain (Subject Editor)
openscience@royalsociety.org

Reviewer comments to Author:

Reviewer: 1

Comments to the Author(s)

This manuscript is clear and convincing. It was meticulously written, with great attention to detail, organized understandably. The statistical analysis is powerful and appropriate. The manuscript is an important contribution to the literature, resolving the issue of whether human death rates level off at advanced ages or continue to increase: the rates level off.

Reviewer: 2

Comments to the Author(s)

The authors focus on lifespans of Italian and French semi-supercentenarians by taking a fully parametric approach based on the Pareto distribution. The main conclusion is that a constant force of mortality describes the data well. While this result is already known for the Italian population, it is here - and for the first time - extended to the French population. The statistical methods are generally correct and the material is nicely described. However, I have two major concerns, listed below.

MAJOR ISSUES

1. The goodness of fit of the model is overlooked. I would welcome a comparison between the cumulative hazard function predicted by the model and a Nelson-Aalen nonparametric estimate of the cumulative hazard function.
2. Lifetimes are treated as exceedances above a threshold u , which is chosen by trying several thresholds and then choosing the value that stabilize the parameter estimates. This part of the analysis looks rather empirical and it is not very convincing. The sample size decreases (and the variability of the estimates increases) as u increases and it is not clear what the authors mean by "stable estimates". I'd welcome a more rigorous approach in the selection of u .

===PREPARING YOUR MANUSCRIPT===

===PREPARING YOUR REVISION IN SCHOLARONE===

Author's Response to Decision Letter for (RSOS-202097.R0)

See Appendix A.

Decision letter (RSOS-202097.R1)

Dear Professor Davison,

It is a pleasure to accept your manuscript entitled "Human mortality at extreme age" in its current form for publication in Royal Society Open Science.

on behalf of Professor Andreas Kyprianou (Associate Editor) and Mark Chaplain (Subject Editor)
openscience@royalsociety.org

Appendix A

Replies to the reviewer comments on “Human mortality at extreme age”

Léo R. Belzile, Anthony C. Davison, Holger Rootzén and Dmitrii Zholud

August 26, 2021

We thank the two reviewers for their positive feedback and helpful comments. Below we respond to the remarks of Reviewer 2, and then make an additional comment.

Substantive changes made in response to your remarks are shown in red in the revised paper; some minor infelicities of wording have also been corrected, but are unmarked.

Reply to Reviewer 2

1. *The goodness of fit of the model is overlooked. I would welcome a comparison between the cumulative hazard function predicted by the model and a Nelson–Aalen nonparametric estimate of the cumulative hazard function.*

Thank you for proposing these improvements.

Various graphical goodness-of-fit diagnostics have been proposed for right-censored and left-truncated data (Waller & Turnbull, 1992). Figure 6 of the Supporting Information shows quantile-quantile (QQ) plots and compares the Nelson–Aalen estimate for the *lstat* data with the fitted parametric models, adjusted for left-truncation as in Example IV. 1.8 of Andersen et al. (1993). We refer to this in the paper.

The France 2019 and IDL data are doubly truncated, so the QQ plots are awkward to construct and difficult to interpret. Instead Figure 7 of the Supporting Information compares the nonparametric cumulative hazard estimates (Turnbull, 1976; Shen, 2010) for these datasets with those of the fitted parametric models.

2. *Lifetimes are treated as exceedances above a threshold u , which is chosen by trying several thresholds and then choosing the value that stabilize the parameter estimates. This part of the analysis looks rather empirical and it is not very convincing. The sample size decreases (and the variability of the estimates increases) as u increases and it is not clear what the authors mean by “stable estimates”. I’d welcome a more rigorous approach in the selection of u .*

Threshold selection is complicated as there is no “correct” threshold: as you write, the choice of u involves a bias-variance tradeoff (too low a threshold may make the generalized Pareto model approximation poor and extrapolation unreliable, but higher thresholds increase variability and thus reduce the value of the fitted model). The (unfortunately inconclusive) literature on threshold

	gen. Pareto	exponential
Istat	0.47	0.61
France	0.44	0.55
IDL	0.77	0.77

Table 1: P -values for the likelihood ratio test comparing the generalized Pareto and exponential models (null hypothesis) with thresholds 108 (Istat, France 2019) and 110 years (IDL) against the piecewise generalized Pareto model of Northrop & Coleman (2014) with five thresholds at the 0, 0.2, \dots , 0.8 quantiles of the exceedances. The p -values are based on the asymptotic χ^2 distribution.

choice in extreme-value statistics up to 2012 is reviewed by Scarrott & MacDonald (2012). There have since been further proposals (e.g., Wadsworth & Tawn, 2012; Northrop & Coleman, 2014; Lee et al., 2015; Wadsworth, 2016), but each has its drawbacks and none is suitable ‘as is’ for censored and truncated data. Thus the use of stability plots remains standard, despite their drawbacks and informal nature.

For a more formal approach we adapted the likelihood ratio testing approach of Northrop & Coleman (2014) to handle censoring and truncation. The results, given in Table 1 of the present document and in Table 3 of the Supporting Information, support the chosen thresholds.

Additional changes

We have taken advantage of the delay since the paper was first submitted to use the most recent, August 2021, release of the IDL; the Swiss, Australian and Italian data (50 records in all) have been removed, but 234 records added. To make our work as reproducible as possible we now provide a supplementary document with instructions on how to download and preprocess the data we use. This update does not affect our overall conclusions, but the increased sample size leads to narrower confidence intervals for the parameter estimates and higher power to detect finite lifespan.

We also took the opportunity to deal with an issue concerning the truncation bounds for the French data: these records are doubly interval-truncated because the lower bounds of the sampling windows differ for semisupercentenarians and supercentenarians. In order to simplify the treatment and exposition, we now impose a common window (death in 1987–2016). This leads us to discard 41 out of 9853 semi-supercentenarians, of which only six lifetimes exceed 108 years. This removal of $\sim 0.5\%$ of the exceedances above 108 years has no discernible impact on our results or conclusions.

References

- Andersen, P., Borgan, O., Gill, R., & Keiding, N. (1993). *Statistical Models Based on Counting Processes*. New York: Springer Verlag.
- Lee, J., Fan, Y., & Sisson, S. (2015). Bayesian threshold selection for extremal models using measures of surprise. *Computational Statistics & Data Analysis*, 85, 84–99.

- Northrop, P. J. & Coleman, C. L. (2014). Improved threshold diagnostic plots for extreme value analyses. *Extremes*, 17(2), 289–303.
- Scarrott, C. & MacDonald, A. (2012). A review of extreme value threshold estimation and uncertainty quantification. *Revstat – Statistical Journal*, 10(1), 33–60.
- Shen, P.-S. (2010). Nonparametric analysis of doubly truncated data. *Annals of the Institute of Statistical Mathematics*, 62, 835–853.
- Turnbull, B. W. (1976). The empirical distribution function with arbitrarily grouped, censored and truncated data. *Journal of the Royal Statistical Society, Series B*, 38, 290–295.
- Wadsworth, J. L. (2016). Exploiting structure of maximum likelihood estimators for extreme value threshold selection. *Technometrics*, 58(1), 116–126.
- Wadsworth, J. L. & Tawn, J. A. (2012). Likelihood-based procedures for threshold diagnostics and uncertainty in extreme value modelling. *Journal of the Royal Statistical Society: Series B (Statistical Methodology)*, 74(3), 543–567.
- Waller, L. A. & Turnbull, B. W. (1992). Probability plotting with censored data. *American Statistician*, 46(1), 5–12.